# The relationships between screen exposure, parent-child interactions and comprehension in 8-month-old infants: The mediating role of shared viewing and parent-child conversation

**Kexin Tu[1], Chengwei Shen[2], Yan Luo**  **[1,2]\*, Yushi Mo[2], Lanying Jian[3], Xinjie Mei[1], Qiong Zhang[3], Lifang Jin[1], Huiling Qin[3]**

1 College of Medical Humanities, Guizhou Medical University, Guiyang, Guizhou, China, 2 Department of Child Health Care, Guiyang Maternal and Child Health Care Hospital, Guiyang, Guizhou, China, 3 College of Public Health, Guizhou Medical University, Guiyang, Guizhou, China

\* Luoyan_15@163.com

## Abstract

### Objective

To explore the relationships between screen exposure, parent-child interactions and comprehension in 8-month-old infants, and to examine whether shared viewing and parent-child conversation during screen exposure may play mediating role in that relationships.

### Methods

The sample included 437 infants aged 8 months from the Children's Health Department of Guiyang Maternal and Child Health Hospital during January 2022 to February 2023. The use of electronic screen devices was assessed using a screen exposure questionnaire. The Brigance Parent-child interactions Scale was used to assess parent-child interactions and the Putonghua Communicative Development Inventory (PCDI) scale was used to assess infants' word comprehension.

### Results

48.7% of infants were found to be using screens 1–2 days per week. There was a significant difference ($p < 0.05$) in the PCDI-comprehension scores of screen-exposed infants compared to non-screen-exposed infants. Shared viewing and parent-child conversation during screen exposure were positively associated with parent-child interactions ($p < 0.05$). Mediation analysis revealed that parent-child conversation fully mediated between screen exposure and PCDI-comprehension, but partially mediated between parent-child interactions and PCDI-comprehension.

### Conclusions

Shared viewing and parent-child conversation during screen exposure may mediate between screen exposure and comprehension development. Shared viewing, parent-child conversation and parent-child interactions may be protective factors for screen exposure in

**Data Availability Statement:** All the relevant data can be found in the file with the supporting information.

**Funding:** This work was supported by grants from the Health Commission of Guizhou Province (gzwkj2022-199). YL is the key person who received the grant. https://wjw.guizhou.gov.cn/xwzx/tzgg/202112/t20211222_78914877.html Study sponsors had no role in the study design; the collection, analysis, and interpretation of data; the writing of the report; or the decision to submit the manuscript for publication.

**Competing interests:** The authors have declared that no competing interests exist.

comprehension development. Suggests that parents should accompany and communicate with their children when they use electronic screen devices to reduce the negative impact of screen exposure on children's comprehension.

## Introduction

Word comprehension is the process of acquiring information through perception of words, and it is one of the skills mastered by infants in the earliest stages of language learning. [1, 2]. Recent research indicates that infants begin to comprehend words at 6–9 months [1]. Early word learning can be boosted by comprehension [3], studies have found that parents' use of spatial language and gestures, as well as home musical environment, can predict infants' comprehension. [4, 5]. Therefore, exploring other influencing factors on infants' comprehension may have important implications for children's language development.

In modern life, electronic devices have become indispensable, with televisions, mobile phones, computers, and other screen media being prevalent in every household. In particular, portable mobile devices like smartphones and tablets have significantly increased the opportunities for infants and young children to interact with screens [6, 7]. Children's exposure to screens is now increasingly common and continues to trend towards younger ages [8]. In China, children's exposure to screens has increased dramatically [9].

Children's exposure to screen has raised concerns for many reasons [10], Studies have shown that screen exposure during early life not only has negative impacts on vision, sleep and weight, but is also negatively correlated with language development [11–15]. However, a recent meta-analysis has found that screen exposure does not have a negative impact on children's vocabulary, in experimental research, there is actually a positive correlation between screen exposure and vocabulary size [16]. Additionally, there is currently no research examining whether infants' comprehension is affected by screen exposure. Therefore, in the present study, we examined the following first research hypothesis: (1) Screen exposure may have positive impact on infants' comprehension directly.

Early childhood is a critical phase for the acquisition of language skills such as phonetics, comprehension and vocabulary [17]. Parent-child interactions are essential in promoting early childhood development, especially in language skills [18–21]. The quantity and quality of parent-child interaction is an important factor in the development of CDI-comprehension and production in children aged 0 to 2 years [22]. Therefore, in the present study, we examined the following second research hypothesis: (2) Parent-child interactions can enhance infants' comprehension directly.

Children using electronic screen devices on their own may replace opportunities for children and parents to interact and communicate, potentially having indirect negative effects on language development [23]. The recent meta-analysis by Madigan et al. found that shared viewing during screen exposure was associated with improved language skills in children [24]. Moreover, media verbal interactions between mothers and 6-month-old infants during media exposure can moderate the detrimental effects of media exposure on 14-month-old toddlers, and potentially have positive impacts on language development [25].

Therefore, it remains to be further explored whether shared viewing and parent-child conversation during shared viewing (Referencing Shah et al.'s research, the present study defines parent-child conversation during the shared viewing as parent-child conversation [26]) will have an impact on infants' comprehension development.

In the light of the findings mentioned above, the following third and fourth research hypothesis were examined in this study: (3) Parent-child interactions can affect infants' comprehension indirectly through shared viewing and parent-child conversation as an intermediary variable. (4) Screen exposure can affect infants' comprehension indirectly through shared viewing and parent-child conversation as an intermediary variable.

In addition, most of the existing studies were retrospective, and there is limited research specifically focusing on screen exposure of infants. Therefore, this study aims to describe the current status of screen exposure among 8-month-old infants and analyze the relationships between screen exposure, parent-child interactions and comprehension.

## Materials and methods

### Ethical approval

The study was approved by the Institutional Review Board, Guiyang Maternal and Child Health Care Hospital Guiyang Children's Hospital (No. 2021–65). Guardians were informed of the purpose and procedures of the study and were assured of anonymity. Written informed consent was obtained from all parents or legal guardians.

### Sample

The present analysis included 441 infants from 3 January 2022 to 28 February 2023 who underwent routine physical examinations at the Children's Health Department of the Guiyang Maternal and Child Health Care Hospital, the refusal rate was less than 3% [27]. Inclusion criteria were intention to receive pediatric primary care in our department for at least 3 years, infants without hearing or visual impairments, and no diagnosed congenital diseases. The general information questionnaire, the screen exposure questionnaire and the Brigance Parent-child interactions Scale were completed online. The Putonghua Communicative Development Inventory scores were obtained through interviews with parents. After excluding irregular and incomplete questionnaires, A total of 437 (238 males and 199 females) participants were retained in the study.

### Research tools

**General information questionnaire.**   The general information questionnaire collected basic demographic information about the infants and their parents, including the infant's birth weight, history of asphyxia, mode of delivery, pregnancy complications, parental education level, family structure and socioeconomic status, as well as family history of mental illness.

**Screen exposure questionnaire.**   We prepared a screen exposure questionnaire based on the screen time recommendations by the American Academy of Pediatrics [8] (AAP) and the studies of Wu et al. [28] and Klakk et al. [29]. The questionnaire consisted of four parts, the first part encompassed questions on screen types and contents [30] (Multiple-choice items, parents were asked, "*What types of electronic screen devices has your child been exposed to?*"), the second part encompassed questions on frequency of screen exposure (Parents were asked, "*How many hours a day does your child watch electronic screen devices?*"), the third part encompassed questions on parental intents of screen exposure (Multiple-choice items, parents were asked, "*When would you allow your child to view an electronic screen device?*"), and the fourth part encompassed questions on caregiver's behavior during viewing (shared viewing and parent-child conversation were determined from this part, parents were asked, "*How often do you watch electronic screen devices with your children when they are exposed to them?*", "*When you and your child watch an electronic screen device together, how often do you talk with your child*

*about the screen contents*?" Responses were coded categorically as 1 = never, 5 = always). Cronbach's alpha for parts 2 and 4 of the questionnaire ranged from 0.896 to 0.912 [31]. Based on the AAP's guidelines for media usage, no screen time for children under 2, and less than 2 hours per day for children 2 to 12 [32]. The infants in this study were therefore categorized into a screen-exposed group and a non-screen-exposed group [33] (the first item of the questionnaire assessed whether infants had been exposed to electronic screens, and the psychometric properties of the questionnaire were measured by individual items without calculating a composite score).

**Brigance Parent-child interactions Scale (BPCIS).**   The BPCIS developed by Frances Page Glasco was used to test the effect of parent-child interactions [34]. This instrument consists 18 items designed to assess parenting behaviors and perceptions about their children [35]. The questionnaire included item such as "*I play with my child and show him or her things about toys.*" adapted version of the BPPCIS (5-point Likert scale) was tested for reliability (Internal consistency reliability = 0.794) and validity (Kaiser-Meyer-Olkin value = 0.777 and Bartlett's test p-value < 0.05) in Chinese families by Jingjing [36]. The Chinese version of the BPCIS adapted by Jingjing was used in the present study to assess the quality of parent-child interactions [36].

**Putonghua Communicative Development Inventory (PCDI).**   The infant language comprehension development was assessed using the PCDI, this is the Chinese version of the MacArthur-Bat Communicative Development Inventory (MCDI) [37]. The PCDI, which consists of infant and toddler forms [38], is a valid tool to assess comprehension and production language skills in infants and young children [39]. In the present study, 8-month-old infants' word comprehension was assessed by the infant form (Words and Gestures, W&G) [39].

## Quality control

All staff involved in the survey received uniform training, which included instructions on the use of the research instruments and its items. Infant growth and development were assessed by qualified staff with medical training, while other questionnaires were completed by primary caregiver, with guidance and explanation from qualified staff with medical training.

## Statistical analysis

Data was entered in excel and imported into spss version 23.0 (IBM, USA) and jamovi 2.3.12 statistical software for data analysis. The discrete data were described using absolute frequencies and proportions and were analyzed using chi-square tests. Normally distributed continuous data were expressed as mean plus-or-minus standard deviation and independent samples t-test for group comparison; non-normally distributed continuous data were expressed as median (p25, p75) and Mann-Whitney U-test for comparison of two groups, Kruskal-Wallis test for multiple comparisons between groups. Correlations were analyzed using Spearman's correlation coefficient, Hierarchical multiple regressions and mediating effects tests were used to analyze the factors influencing infants' comprehension, $p < 0.05$ was considered significant.

## Results

### Descriptive and correlational analyses of the study variables

The 437 infants were divided into screen-exposed group (320, 73.2%) and non-screen-exposed group (117, 26.8%) according to whether they had viewed a screen or not. Smartphones (91.6%) were the most common type of electronic screen device used by infants in the screen-exposed group, with 90 (28.1%) of them having been exposed to a smartphone by 5 months of

**Table 1. The general information of the sample.**

|  | non-screen-exposed group | screen-exposed group |  |  |
| --- | --- | --- | --- | --- |
| Gender |  |  | 0.876 | 0.024 |
| Male | 63 (53.8%) | 175 (54.7%) |  |  |
| Female | 54 (46.2%) | 145 (55.3%) |  |  |
| High risk infants |  |  | 0.369 | 0.806 |
| No | 51 (43.6%) | 155(35.9%) |  |  |
| Yes | 66 (56.4%) | 165(64.1%) |  |  |
| Mother's age | 32.9±4.32 | 32.4±3.62 | 0.690 | 0.159 |
| Father's age | 33±4.27 | 32.3±3.79 | 0.467 | 0.529 |
| Mother's education level |  |  | 0.203 | 1.621 |
| Level 1 | 5 (4.2%) | 22 (6.9%) |  |  |
| Level 2 | 98 (83.8%) | 269 (84.1%) |  |  |
| Level 3 | 14 (12%) | 29 (9.%) |  |  |
| Father's education level |  |  | 0.059 | 3.561 |
| Level 1 | 4 (3.4%) | 16 (5%) |  |  |
| Level 2 | 96 (82.1%) | 277 (86.6%) |  |  |
| Level 3 | 17 (14.5%) | 27 (8.4%) |  |  |
| family structure |  |  | 0.106 | 2.620 |
| Single parent family | 3 (2.6%) | 4 (1.3%) |  |  |
| Joint family | 4 (3.4%) | 11 (3.4%) |  |  |
| Immediate family | 51 (43.6%) | 116 (36.3%) |  |  |
| Nuclear family | 59 (50.4%) | 189 (59%) |  |  |
| Self-reported household income |  |  | 0.002 | 9.906 |
| ≤4999 RMB/month | 1 (0.9%) | 19 (6%) |  |  |
| 5000–9999 RMB/month | 90 (76.9%) | 261 (81.5%) |  |  |
| ≥10000 RMB/month | 26 (22.2%) | 40 (12.5%) |  |  |
| parental time with the child(h/day) | 4.74±3.71 | 5.46±4.42 | 0.048 | 3.895 |

Note. Levels of education: 1—high school and below, 2—college and university, 3—postgraduate and above

age. 176 (55%) parents reported that they would allow their child access to screen when using electronic screen device themselves. The differences between the screen-exposed group and non-screen-exposed group were statistically significant in terms of mean monthly income ($\chi^2$ = 9.906, $p$ = 0.002) and mean time parents spent with their children per day ($\chi^2$ = 3.895, $p$ = 0.048). The general information of the sample is shown in Table 1.

289 infants viewed the screen 1–2 times per day (66.1%). 282 infants had screen time of less than 30 minutes per viewing (64.5%). 177 parents did not watch screens with their children, while 240 parents watched screens with their children and talked about the screen contents. The screen usage of the infants in the screen-exposed group is presented in Fig 1. On PCDI-comprehension, there were significant differences in whether infants viewed screens ($U$ = 15343.000, $p$ = 0.004), screen time per viewing ($\chi^2$ = 9.572, $p$ = 0.023), frequency of screen exposure per day ($\chi^2$ = 9.635, $p$ = 0.022), the number of days of screen exposure per week($\chi^2$ = 11.112, $p$ = 0.011), shared viewing ($\chi^2$ = 14.617, $p$ = 0.012) and parent-child conversation ($\chi^2$ = 21.936, $p < 0.001$) (Table 2). On the parent-child interactions scores, there were significant differences in screen time per viewing ($\chi^2$ = 8.792, $p$ = 0.032) and parent-child conversation ($\chi^2$ = 22.672, $p < 0.001$) (Table 2).

Correlation analysis showed that screen exposure was not significantly correlated with parent-child interactions ($r$ = 0.015, $p$ = 0.749), but screen exposure was significantly correlated

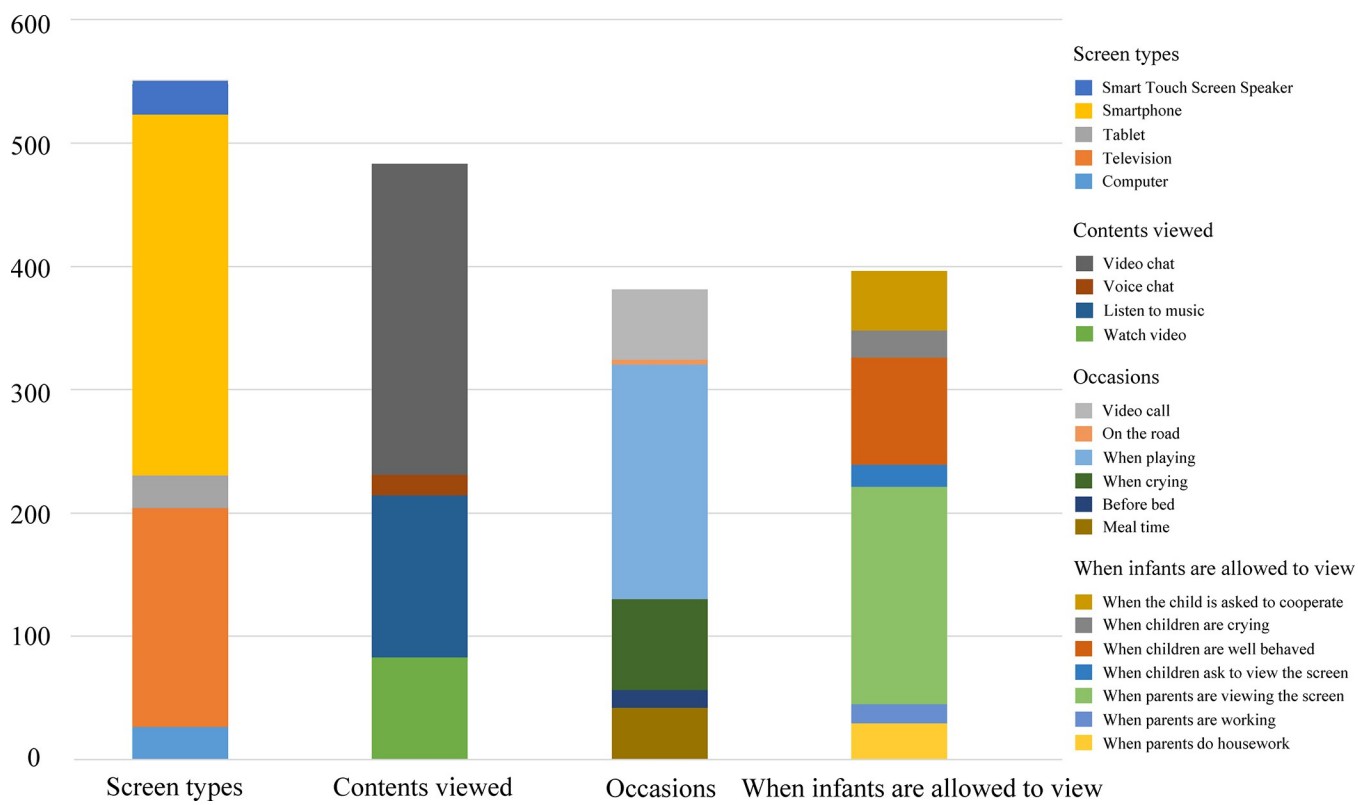

**Fig 1. The screen usage of the infants in the screen-exposed group.** screen types, contents viewed, occasions and when infants are allowed to view are all multiple-choice items. The Y-axis represents the cumulative number of times that the item has been selected.

with PCDI-comprehension ($r = 0.139$, $p = 0.004$). Parent-child interactions was significantly correlated with PCDI-comprehension ($r = 0.142$, $p = 0.003$) and parent-child conversation ($r = 0.108$, $p = 0.023$) (Table 3).

## Regression and mediating effects analysis

Hierarchical multiple regression was adopted to investigate the impact that socio-demographic details, parental time with the child, parent-child interactions, and screen exposure on the PCDI-comprehension [40]. A total of three models had been adopted: model 1 was inclusive of socio-demographic variables (gender, parental education level, household income); model 2 added parental time with the child; and model 3 added parent-child interactions and screen exposure. The multicollinearity analyses showed that there were little interactions between the factors (all variance inflatable factors were less than 2), There were no signs of multicollinearity in any of the three regression models (Table 4).

The PCDI-comprehension predictors can be found in Table 4, and model 1 indicated that socio-demographic variables could only explain 1.3% of the PCDI-comprehension variance ($p > 0.05$). After parental time with the child was added to model 2, its overall explanatory power had been increased to 2.2% ($p > 0.05$), parental time with the child had a borderline significant impact on PCDI- comprehension ($\beta = 0.094$, $p = 0.05$). Model 3 had again added parent-child interactions and screen exposure for the predictive variable. The explanatory power of this model was 5.4%, with an increase of 3.2%, thereby showing that parent-child

**Table 2. Differences between parent-child interactions and PCDI comprehension in screen-exposed infants aged 8 months.**

| Variable | Cases (n/%) | Parent-child interactions | $\chi^2/U$ | PCDI-comprehension | $\chi^2/U$ |
|---|---|---|---|---|---|
| screen exposure | | | 18345.000a | | 15343.000a** |
| No | 117(26.8%) | 73(66,79) | | 34(12,46) | |
| Yes | 320(73.2%) | 73.5(68,79) | | 39(19,58) | |
| days/week | | | 0.250b | | 11.112b* |
| No use | 117(26.8%) | 73(66,79) | | 34(12,46) | |
| ≤2days | 213(48.7%) | 73(68,79) | | 39(21,59) | |
| 3-5days | 81(18.5%) | 74(68,80) | | 39(23,60) | |
| ≥6days | 26(6%) | 72(67,78.8) | | 26.5(14,52.3) | |
| times/day | | | 1.974b | | 9.635b* |
| no exposure | 117(26.8%) | 73(66,79) | | 34(12,46) | |
| 1-2times | 289(66.1%) | 73(68,79) | | 39(18.3,58) | |
| 3-4imes | 24(5.5%) | 76(70,82.8) | | 42(29.3,62.5) | |
| ≥5times | 7(1.6%) | 64(62.5,66.5) | | 39(30.8,44.3) | |
| h/time | | | 8.792b* | | 9.572b* |
| no exposure | 117(26.8%) | 73(66,79) | | 34(12,46) | |
| <0.5h | 282(64.5%) | 73(68,79) | | 39(18.3,58) | |
| 0.5-1h | 34(7.8%) | 76(70,82.8) | | 42(29.3,62.5) | |
| >1h | 4(0.9%) | 64(62.5,66.5) | | 39(30.8,44.3) | |
| shared viewing | | | 5.214b | | 14.617b** |
| no exposure | 117(26.8%) | 73(66,79) | | 34(12,46) | |
| never | 60(13.7%) | 74.5(68.8,77.5) | | 39(24.8,45.3) | |
| occasionally | 141(32.3%) | 74(68,79) | | 39(19,58) | |
| sometimes | 83(19%) | 71(65.5,79.5) | | 39(14,61) | |
| often | 24(5.5%) | 74(68.5,78.3) | | 55.5(34.8,85.8) | |
| always | 12(2.7%) | 78.5(74,83.5) | | 35(19.3,76.3) | |
| parent-child conversation | | | 22.672b*** | | 21.936b*** |
| no exposure | 117(26.8%) | 73(66,79) | | 34(12,46) | |
| never | 80(18.3%) | 72(65.8,77) | | 39(25.5,44.8) | |
| occasionally | 94(21.5%) | 74(68,77.8) | | 38.5(16,56.5) | |
| sometimes | 91(20.8%) | 71(65,79) | | 39(14.5,63) | |
| often | 31(7.1%) | 76(74,79) | | 63(39.5,85) | |
| always | 24(5.5%) | 80(75,83) | | 39(19.3,58.5) | |

Note. a: Mann-Whitney U test, b: Kruskal-Wallis test

*P<0.05

**P<0.01

***P<0.001

interactions ($\beta = 0.142$, $p = 0.003$) and screen exposure ($\beta = 0.113$, $p = 0.018$) could significantly explain PCDI-comprehension (Table 4).

Through mediating effects analysis, we found that shared viewing ($\beta = 0.135$, 95% $CI = 1.420$–15.299, $p_m = 0.018$) was a full mediator in the relationship between screen exposure and PCDI-comprehension (Table 5 and Fig 2), and parent-child conversation was a full mediator in the relationship between screen exposure and PCDI-comprehension ($\beta = 0.118$, 95% $CI = 1.087$–13.487, $p_m = 0.021$) and parent-child conversation was a partial mediator in the relationship between parent-child interactions and PCDI- comprehension ($\beta = 0.020$, 95% $CI = 15.350$–0.124, $p_m = 0.047$) (Table 6 and Fig 3).

**Table 3. The Correlation analysis of screen exposure, parent-child interactions, and l PCDI comprehension in 8-month-old infants.**

| Variable | 1 | 2 | 3 | 4 | 5 | 6 | 7 | 8 | 9 | 10 |
|---|---|---|---|---|---|---|---|---|---|---|
| 1.parent-child interaction | — | | | | | | | | | |
| 2.PCDI-comprehension | 0.142** | — | | | | | | | | |
| 3. interaction time (h/day) | 0.176*** | 0.106* | — | | | | | | | |
| 4.screen exposure | 0.015 | 0.139** | 0.095* | — | | | | | | |
| 5.day/week | 0.013 | 0.101* | 0.080 | 0.828*** | — | | | | | |
| 6.time/day | -0.009 | 0.108* | 0.066 | 0.922*** | 0.847*** | — | | | | |
| 7.h/time | 0.041 | 0.143** | 0.039 | 0.909*** | 0.838*** | 0.861*** | — | | | |
| 8.shared viewing | 0.006 | 0.161*** | 0.111* | 0.792*** | 0.744*** | 0.778*** | 0.773*** | — | | |
| 9.parent-child conversation | 0.108* | 0.183*** | 0.126** | 0.785*** | 0.725*** | 0.755*** | 0.771*** | 0.838*** | — | |
| 10.video chat time | 0.026 | 0.104* | 0.080 | 0.837*** | 0.762*** | 0.807*** | 0.821*** | 0.696*** | 0.696*** | — |

Note
*P<0.05
** P<0.01
***P<0.001

## Discussion

The present study found that 73.2% of 8-month-old infants had been exposed to electronic screen devices, 55% had been exposed to electronic screens before the age of 5 months, and that smartphones and televisions were the most common types of electronic screen devices that infants were exposed to. 65% of infants used electronic screens 1–2 times per day, with an average of less than half an hour per viewing, the most common use of electronic screen devices by infants was for video chatting with other family members and relatives, shared viewing and parent-child conversation mediated the relationship between screen exposure and PCDI-comprehension.

**Table 4. Predictors of PCDI- comprehension: Hierarchical multiple regressions.**

| | Model 1 | | | | Model 2 | | | | Model 3 | | | |
|---|---|---|---|---|---|---|---|---|---|---|---|---|
| | *β* | *t* | *P* | *VIF* | *β* | *t* | *P* | *VIF* | *β* | *t* | *P* | *VIF* |
| Gender | .046 | .955 | .340 | 1.007 | .047 | .975 | .330 | 1.007 | .052 | 1.112 | .267 | 1.008 |
| High risk infants | .076 | 1.588 | .113 | 1.014 | .075 | 1.560 | .120 | 1.014 | .079 | 1.676 | .094 | 1.017 |
| Mother's education level | .067 | 1.256 | .210 | 1.243 | .074 | 1.391 | .165 | 1.249 | .061 | 1.150 | .251 | 1.259 |
| Father's education level | -.023 | -.422 | .674 | 1.259 | -.017 | -.319 | .750 | 1.262 | -.020 | -.385 | .701 | 1.269 |
| Self-reported household income | -.039 | -.773 | .440 | 1.085 | -.039 | -.791 | .429 | 1.085 | -.029 | -.587 | .558 | 1.109 |
| parental time with the child(h/day) | | | | | .094 | 1.961 | .050 | 1.013 | .065 | 1.359 | .175 | 1.041 |
| parent-child interactions | | | | | | | | | .142 | 2.961 | .003 | 1.044 |
| screen exposure | | | | | | | | | .113 | 2.369 | .018 | 1.036 |
| $R^2$ | .013 | | | | .022 | | | | .054** | | | |
| Adjusted $R^2$ | .002 | | | | .008 | | | | .037** | | | |
| $R^2$ change | .013 | | | | .009 | | | | .032** | | | |
| *F* change | 1.164 | | | | 3.847 | | | | 7.277** | | | |

Note
*P<0.05
** P<0.01
***P<0.001

**Table 5. Results of the mediation analyses of screen exposure, parent-child interactions, shared viewing and PCDI-comprehension.**

| | effect | SE | β | Z | P | OR(95%CI) |
|---|---|---|---|---|---|---|
| indirect | screen exposure ⇒ PCDI-comprehension | 3.541 | 0.135 | 2.361 | 0.018 | 1.420~15.299 |
| | parent-child interactions ⇒ shared viewing ⇒ PCDI-comprehension | 0.017 | 0.002 | 0.294 | 0.769 | -0.028~0.038 |
| component | screen exposure ⇒ shared viewing | 0.091 | 0.775 | 25.605 | < .001 | 2.156~2.513 |
| | screen exposure⇒ PCDI-comprehension | 1.510 | 0.175 | 2.371 | 0.018 | 0.621~6.541 |
| | parent-child interactions ⇒ shared viewing | 0.005 | 0.009 | 0.296 | 0.767 | -0.008~0.011 |
| direct | screen exposure ⇒ PCDI-comprehension | 4.551 | -0.019 | -0.251 | 0.802 | -10.063~7.777 |
| | parent-child interactions ⇒ PCDI-comprehension | 0.147 | 0.152 | 3.257 | 0.001 | 0.191~0.766 |
| total | screen exposure ⇒ PCDI-comprehension | 2.900 | 0.117 | 2.489 | 0.013 | 1.533~12.901 |
| | parent-child interactions ⇒ PCDI-comprehension | 0.148 | 0.153 | 3.267 | 0.001 | 0.193~0.774 |

Previous research has found that the majority of households with children aged 6 months to 4 years have mobile or fixed screens, including televisions (97%), tablets (83%) and smartphones (77%), furthermore, 96.6% of children start using electronic screens before the age of 1 [41]. Zimmerman et al. showed a significant negative correlation between time spent watching videos and PCDI scores for children aged 8 months to 16 months [42]. Shi et al. showed that children with delayed language development were first exposed to screens at 11.6 months of age and spent an average of 118.8 minutes per day on screens, younger age of first screen exposure and longer average daily screen time were associated with reduced parent-child communication and lower levels of language development in children [43]. However, research of highly educated parents found no negative effects of screen exposure on children's vocabulary, and Taylor et al. suggested that moderate media use will not adversely affect children's vocabulary development as long as family reading time is not reduced by screen exposure [44].

Recently, a growing body of research suggests that exposure to electronic screens has no negative impact on children's vocabulary and may even promote early vocabulary development [16, 24]. This may be related to the media contents and the parent-child interactions during screen time [33, 45]. Linebarger et al.'s study found that children's vocabulary growth was related to the types of screen content [46], and that the infants in the present study were exposed to relatively benign digital media (consisting mainly of video/voice chat and listening to music). Parent-child interactions during screen exposure may be a new form of early childhood education, and such interactions may moderate the impacts of screen exposure by reducing adverse effects and increasing the possibility of benefits [47]. When children are exposed to electronic screen devices, parents can improve their comprehension by co-viewing and

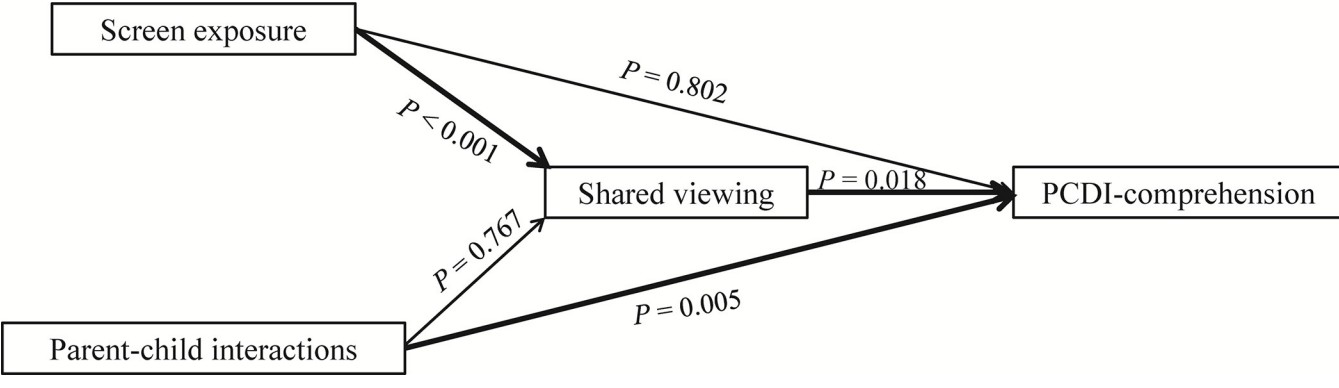

**Fig 2. Observed relationships between screen exposure, parent-child interactions, shared viewing and PCDI-comprehension.**

**Table 6. Results of the mediation analyses of screen exposure, parent-child interactions, parent-child conversation and PCDI-comprehension.**

| | effect | *SE* | *β* | *Z* | *P* | *OR(95%CI)* |
|---|---|---|---|---|---|---|
| indirect | screen exposure ⇒ parent-child conversation ⇒ PCDI-comprehension | 3.163 | 0.118 | 2.304 | 0.021 | 1.087~13.487 |
| | parent-child interactions ⇒ parent-child conversation ⇒ PCDI-comprehension | 0.032 | 0.020 | 1.983 | 0.047 | 15.350~0.124 |
| component | screen exposure ⇒ parent-child conversation | 0.107 | 0.732 | 22.853 | < .001 | 2.241~2.662 |
| | parent-child conversation ⇒ PCDI-comprehension | 1.284 | 0.161 | 2.315 | 0.021 | 0.456~5.489 |
| | parent-child interactions ⇒ parent-child conversation | 0.005 | 0.123 | 3.838 | < .001 | 0.010~0.032 |
| direct | screen exposure ⇒ PCDI-comprehension | 4.266 | -0.001 | -0.017 | 0.987 | -8.431~8.290 |
| | parent-child interactions ⇒ PCDI-comprehension | 0.149 | 0.134 | 2.818 | 0.005 | 0.128~0.714 |
| total | screen exposure ⇒ PCDI-comprehension | 2.900 | 0.117 | 2.489 | 0.013 | 1.533~12.901 |
| | parent-child interactions ⇒ PCDI-comprehension | 0.148 | 0.153 | 3.267 | 0.001 | 0.193~0.774 |

engaging in conversation about the contents on the screen. However, this may require benign screen contents as a prerequisite [24, 39, 48].

Comprehension is an important component of early language development in infants [2]. The mediation effect of shared viewing and parent-child conversation in the relationships between screen exposure, parent-child interactions and PCDI- comprehension in the present study seems to be a new finding. The results of the mediating effects analysis may explain why infants in the screen-exposed group had higher language comprehension than the non-exposed group. As infants have difficulty viewing screens alone, parent-child conversation during shared viewing may potentially increase the duration of parent-child interactions, thereby promoting infants' comprehension. Shared viewing did not play a mediating role in parent-child interactions and comprehension, but parent-child conversation partially mediated this relationship, parent-child conversation may serve as one of the ways for parent-child interactions during screen exposure. Therefore, in order to enhance infants' comprehension and provide a language-rich environment during shared screen time, parents need to actively interact with their infants and verbally describe the contents on screens [49].

Children's exposure to electronic screens has become commonplace [49], and the AAP recommends that children younger than 18 months of age should avoid screen exposure [8]. Although many parents are aware of the recommendations, adherence varies widely in actual parental behavior [32]. Given the large number of children prematurely exposed to electronic

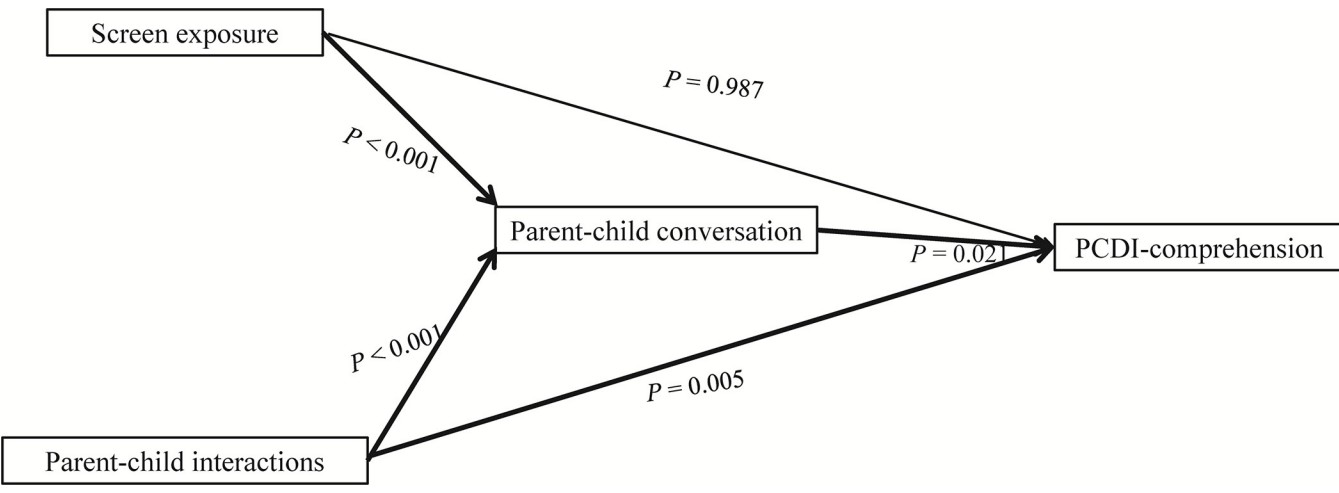

**Fig 3. Observed relationships between screen exposure, parent-child interactions, parent-child conversation and PCDI-comprehension.**

screen devices, parent-child conversation during shared viewing may play a protective role in language development of screen-exposed children [25, 26, 33, 44].

## Limitations

There are some limitations in the present study. (1) This is a cross-sectional study, and subjects were 8-month-old infants, further prospective cohort studies are needed to explore the causal relationships between screen exposure, parent-child interactions, and language development at different ages. (2) The sample size was small, with only 437 participants. (3) We did not investigate the relationships between screen contents and infants' comprehension, and future research could explore the long-term effects of screen contents on children's language development. (4) The questionnaire for screen exposure in the present study was parent-reported. Barr et al. argue that comprehensive and systematic measurement tools of early media exposure are needed [50]. Therefore, future research could use tracking applications on electronic screen devices to objectively monitor screen use, such as the Comprehensive Assessment of Courtroom Media Exposure (CAFE) Consortium [50–52].

## Supporting information

**S1 File. Raw data.**
(XLSX)

**S2 File. Screen exposure questionnaire.**
(DOCX)

## Author Contributions

**Conceptualization:** Chengwei Shen, Yan Luo, Yushi Mo.

**Data curation:** Kexin Tu, Lanying Jian, Xinjie Mei, Qiong Zhang, Lifang Jin, Huiling Qin.

**Formal analysis:** Kexin Tu.

**Supervision:** Yan Luo.

**Writing – original draft:** Kexin Tu.

**Writing – review & editing:** Kexin Tu, Chengwei Shen, Yushi Mo.

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
