## [Decision Letter · Decision Letter 0]

17 Jul 2023

PONE-D-23-17586The relationships between screen exposure, parent-child interactions and comprehension in 8-month-old infants: The mediating role of shared viewing and parent conversationPLOS ONE

Dear Dr. Luo,

Thank you for submitting your manuscript to PLOS ONE. After careful consideration, we feel that it has merit but does not fully meet PLOS ONE’s publication criteria as it currently stands. Therefore, we invite you to submit a revised version of the manuscript that addresses the points raised during the review process.

We look forward to receiving your revised manuscript.

Kind regards,

Anastassia Zabrodskaja, Ph.D.

Academic Editor

PLOS ONE

Journal Requirements:

   "No"

Additional Editor Comments:

Please pay close attention to all reviewers' comments.

Reviewers' comments:

Reviewer's Responses to Questions

**Comments to the Author**

1. Is the manuscript technically sound, and do the data support the conclusions?

Reviewer #1: No

Reviewer #2: Partly

2. Has the statistical analysis been performed appropriately and rigorously? 

Reviewer #1: No

Reviewer #2: Yes

3. Have the authors made all data underlying the findings in their manuscript fully available?

Reviewer #1: No

Reviewer #2: Yes

4. Is the manuscript presented in an intelligible fashion and written in standard English?

Reviewer #1: No

Reviewer #2: Yes

5. Review Comments to the Author

Reviewer #1: I thank the authors for this manuscript, which investigate an interesting and timely topic in the language development literature.

Unfortunately, I have to recommend that the manuscript be rejected in its current form, because of important methodological flaws.

First, no information is provided about the screen exposure scales. We know nothing about the composition and validity of constructs. How is the Shared Viewing construct defined? What about the Parent Conversation construct?

More details are provided on the Brigance Parent-child interactions Scale (BPCIS), but this scale does not seem to be specifically related to interactions during screen time. Also, not only we don't have information about the scale's validity, but the scale was manipulated by the authors (two items were added, and the scale was changed to 5 points).

I am also concerned with the fact that background variables are not included in the analysis as control variables. The two groups — screen-exposed v non-exposed children — differ significantly in relation to the amount of time that the parent spends with the child, with parents in the screen-exposed group spending more time with their child during an average day. However, it's not really clear to me how this variable relates to the Parent Conversation, Parent-Child Interactions, and Shared Viewing variables. For what I can see, the higher CDI scores may very well be driven by the fact that parents in the screen-exposed groups are spending more time in one-to-one situations with an adult (vs. one-to-many). We may want to factor this out: Holding constant the amount of time that parents spend with their children, what is the effect of spending some of this time looking at a screen?

I can't see the rationale behind the authors' decision to divide the children in two exposure groups, instead of analyzing exposure as a continuous variable. I worry whether this dichotomization may have biased the analyses.

I also wonder whether the authors are aware of this new meta-analysis, based on 11,413 children aged 0-6 years, showing that screen time (including non-shared screen time) has no negative effect on vocabulary size and, if anything, it has a small positive effect? The manuscript should be reframed to have less focus on the negative effects of screen exposure.

While it is true that the authors have made their data available, the variable names have not been translated to English, which makes it effectively impossible to use the data set for international researchers.

Reviewer #2: Review PONE-D-23-17586

This a Chinese study reporting on screen exposure in 8 months old infants as reported by over 400 parents. The focus on a very young age group as well as being based on a non-western sample could make it an essential contribution to the field.

There are however a few questions/issues that I believe needs to be explained/addressed before a possible publication.

1. Selection of sample. How many families were approached? As it reads now, not a single family denied participating. If so it needs some explanation. The only attrition mentioned is attributed to incomplete questionnaires (n=4).

2. Ethics. Was the Helsinki convention followed and were the families allowed to withdraw their consent without consequences regarding future care?

3. Method. How long time did the assessment take and was it completed at the clinic? Wad it filled in by the mother, father or conjointly?

4.Quality cntrl. What does “uniform training” entail?

5. Instruments. PCDI. Is there a Chinese validation that has been published? The reference is to a clinical study (25). The same goes for the Brigance scale (23). What was the reason changing the scale from 3- to 5-points?

6. Screen exposure questionnaire. Is it available online or by request? Or is it possible to give an example question? Also: How does it relate to the ongoing debate on how to capture screen time in families with young children (e.g., Barr, R., et al. (2020). Beyond Screen Time: A synergistic approach to a more comprehensive assessment of family media exposure during early childhood, Frontiers in Psychology. 11:1283.)

7. Comprehension score. It is maybe a bit surprising that the children not being exposed to any screens have a lower comprehension score (effect size=?). This find would benefit from one or two more sentences. Could it be that the uneven financial distribution between the groups could be an answer? There is also a near significant effect in the mother’s educational level. Or is it the case that video chat has a positive effect and that this is the effect that the study captures?

8. Content. An improvement of the paper would be to highlight content more. It is almost not mentioned in the background or in the discussion. But content matters! It is obvious from Figure 1 that the screen exposure is relatively benign: The digital media content is almost completely made up of video/voice chat plus listening to music. A rough estimate based on the figure suggests that those aspects make up about 85% of the content. This is not a clear message as the paper is currently written.

9. Limitations: I would not call N = 437 for a small sample but that is of course my subjective opinion. I would prefer “a relatively small sample”.

10. Figures: Figure legends are missing. It is unclear what the Y-axis represents in Fig 1, I have assumed minutes.

6. PLOS authors have the option to publish the peer review history of their article (what does this mean?). If published, this will include your full peer review and any attached files.

Reviewer #1: No

Reviewer #2: No

---

## [Author Response · Author response to Decision Letter 0]

4 Sep 2023

Dear Editors and Reviewers:

Thank you for your letter and for the reviewers’ comments concerning our manuscript entitled “The relationships between screen exposure, parent-child interactions and comprehension in 8-month-old infants: The mediating role of shared viewing and parent-child conversation” (ID: PONE-D-23-17586). Those comments are all valuable and very helpful for revising and improving our paper, as well as the important guiding significance to our researches. We have studied comments carefully and have made correction which we hope meet with approval.

SUGGESTIONS FROM EDITOR

 "No"

Response to editor 

1.We have revised the manuscript according to PLOS ONE's style requirements. 

2.We have removed the funding-related text from the manuscript.

3.The relevant parts of the funding information have been reviewed and amended. 

4.We have stated (in our cover letter) that the funders had no role in study design, data collection and analysis, decision to publish, or preparation of the manuscript. 

5. All relevant data are within the manuscript and its Supporting Information files. And we chose the option “Tick here if your circumstances are not covered by the questions above and you need the journal’s help to make your data available.”

6.The full name of the ethics committee was given in the methods section of the manuscript, and informed written consent was obtained from the participants is also stated in the methods.

Reviewer #1: I thank the authors for this manuscript, which investigate an interesting and timely topic in the language development literature.

Unfortunately, I have to recommend that the manuscript be rejected in its current form, because of important methodological flaws.

First, no information is provided about the screen exposure scales. We know nothing about the composition and validity of constructs. How is the Shared Viewing construct defined? What about the Parent Conversation construct?

More details are provided on the Brigance Parent-child interactions Scale (BPCIS), but this scale does not seem to be specifically related to interactions during screen time. Also, not only we don't have information about the scale's validity, but the scale was manipulated by the authors (two items were added, and the scale was changed to 5 points).

I am also concerned with the fact that background variables are not included in the analysis as control variables. The two groups — screen-exposed v non-exposed children — differ significantly in relation to the amount of time that the parent spends with the child, with parents in the screen-exposed group spending more time with their child during an average day. However, it's not really clear to me how this variable relates to the Parent Conversation, Parent-Child Interactions, and Shared Viewing variables. For what I can see, the higher CDI scores may very well be driven by the fact that parents in the screen-exposed groups are spending more time in one-to-one situations with an adult (vs. one-to-many). We may want to factor this out: Holding constant the amount of time that parents spend with their children, what is the effect of spending some of this time looking at a screen?

I can't see the rationale behind the authors' decision to divide the children in two exposure groups, instead of analyzing exposure as a continuous variable. I worry whether this dichotomization may have biased the analyses.

I also wonder whether the authors are aware of this new meta-analysis, based on 11,413 children aged 0-6 years, showing that screen time (including non-shared screen time) has no negative effect on vocabulary size and, if anything, it has a small positive effect? The manuscript should be reframed to have less focus on the negative effects of screen exposure.

While it is true that the authors have made their data available, the variable names have not been translated to English, which makes it effectively impossible to use the data set for international researchers.

Responds to the reviewer’s comments:

1.Thanks to your comments, we have translated and uploaded the screen exposure questionnaire, and for co-viewing and parental conversation are also defined in the introduction and methods, and examples of relevant items are given in the methodology section.

2. The Chinese version of the BPCIS was based on a study by Jingjing Lai, who changed the BPCIS to a 5-point scale and analyzed the reliability (Internal consistency reliability = 0.794) and validity (Kaiser-Meyer-Olkin value = 0.777 and Bartlett's test p-value < 0.05). Glasco mentions that the BPCIS is more closely related to language development in her research (Glascoe FP. The Brigance Infant and Toddler Screen: standardization and validation. J Dev Behav Pediatr. 2002 Jun;23(3):145-50. doi: 10.1097/00004703-200206000-00003.), So we chose the BPCIS as the instrument to measure the level of parent-child interaction in this study.

Glasco mentions that the BPCIS is more closely related to language development in her research. (Glascoe, 2002),

3. Thank you very much for this important comment, Hierarchical multiple regressions were conducted to identify the predictors of PCDI-comprehension and control for the effect of parental time with the child on PCDI- comprehension. In model 3, we discovered that parent-child interaction and screen exposure could significantly explain PCDI-comprehension. To consult for multicollinearity between the factors, multicollinearity analyses has been used in each step. The results shows that the interactions between the factors were little because all VIF in collinear diagnostics were less than 2. there were no signs of multicollinearity in any of the 3 regression models. More results from the hierarchical regression are reported in the results section of the manuscript. Thank you again for your comments.

4. Thanks for your comments, due to the design of the questionnaire, screen exposure was not measured as a continuous variable. But We divided screen exposure into more levels (the screen time per viewing, the number of screen exposure per day, The number of days of screen exposure per week), exploring the relationship between these different levels of screen exposure and PCDI-understanding, and more about this is described in the results and discussion of the manuscript.

5. Thank you very much for the information about this new meta-analysis, which we have carefully read and revised our introduction and discussion to take account of the content of this meta-analysis.

6. Thanks for your comments, it was an oversight on our part, we have translated the file into English and uploaded it.

Reviewer #2:

This a Chinese study reporting on screen exposure in 8 months old infants as reported by over 400 parents. The focus on a very young age group as well as being based on a non-western sample could make it an essential contribution to the field.

There are however a few questions/issues that I believe needs to be explained/addressed before a possible publication.

1. Selection of sample. How many families were approached? As it reads now, not a single family denied participating. If so it needs some explanation. The only attrition mentioned is attributed to incomplete questionnaires (n=4).

2. Ethics. Was the Helsinki convention followed and were the families allowed to withdraw their consent without consequences regarding future care?

3. Method. How long time did the assessment take and was it completed at the clinic? Wad it filled in by the mother, father or conjointly?

4.Quality cntrl. What does “uniform training” entail?

5. Instruments. PCDI. Is there a Chinese validation that has been published? The reference is to a clinical study (25). The same goes for the Brigance scale (23). What was the reason changing the scale from 3- to 5-points?

6. Screen exposure questionnaire. Is it available online or by request? Or is it possible to give an example question? Also: How does it relate to the ongoing debate on how to capture screen time in families with young children (e.g., Barr, R., et al. (2020). Beyond Screen Time: A synergistic approach to a more comprehensive assessment of family media exposure during early childhood, Frontiers in Psychology. 11:1283.)

7. Comprehension score. It is maybe a bit surprising that the children not being exposed to any screens have a lower comprehension score (effect size=?). This find would benefit from one or two more sentences. Could it be that the uneven financial distribution between the groups could be an answer? There is also a near significant effect in the mother’s educational level. Or is it the case that video chat has a positive effect and that this is the effect that the study captures?

8. Content. An improvement of the paper would be to highlight content more. It is almost not mentioned in the background or in the discussion. But content matters! It is obvious from Figure 1 that the screen exposure is relatively benign: The digital media content is almost completely made up of video/voice chat plus listening to music. A rough estimate based on the figure suggests that those aspects make up about 85% of the content. This is not a clear message as the paper is currently written.

9. Limitations: I would not call N = 437 for a small sample but that is of course my subjective opinion. I would prefer “a relatively small sample”.

10. Figures: Figure legends are missing. It is unclear what the Y-axis represents in Fig 1, I have assumed minutes.

Responds to the reviewer’s comments:

1. Thank you for your comments, some parents refused to participate, but we did not count the number of refusals, and as far as we can recall no more than 10 refused to participate in this study (parents receive routine physical examinations in our department, and generally have sufficient time to complete the questionnaire).

2. We obtained ethical approval from the Guiyang Maternal and Child Health Care Hospital and the study adhered to the Principles of the Declaration of Helsinki. The families allowed to withdraw their consent without consequences regarding future care.

3. All questionnaires were completed by either the father or the mother (the one who knew the child better) and took approximately 20 minutes to complete, with 62.93% of the total sample completing them in the clinic and 37.07% completing them online, again under the guidance of trained medical staff.

4. The uniform training includes knowledge of the questionnaire items and their use. We have also added the explanation of "uniform training" to the quality control section.

5. There is a published Chinese version of the validation of the PCDI (Tardif T et al., 2009, doi:10.1017/S0305000908009185), as well as the standardization and validation of BPCIS (Glascoe et al, 2002 doi:10.1097/00004703 -200206000-00003), and we have added these two papers to the references. The reason for changing the BPCIS from a 3-point scale to a 5-point scale is that we refer to the study of Jingjing Lai, who translated the BPCIS into Chinese and changed it to a 5-point scale，and analyzed the reliability(Internal consistency reliability = 0.794) and validity(Kaiser-Meyer-Olkin value = 0.777 and Bartlett's test p-value < 0.05). (Jingjing L，2013)

6. We have translated and uploaded the questionnaire, and provided sample question in the methods section. Thank you for the information you provided about the paper, we also read it carefully and contributed to the discussion.

7. Thank you for your comments. Regarding the comprehension score, we reviewed previous research and found a correlation between it and a number of factors, including the level of parental education and the level of family income that you mentioned. However, in terms of correlations, our results showed significant correlations between comprehension scores and parent-child interactions, co-viewing, and parent-child conversation during co-viewing, which we also explain in the discussion. 

8. Thanks for your comments, we have made changes to the relevant parts of the discussion in line with your comments.

9. Thanks a lot for your comments.

10. Thank you very much for your comments, the lack of a figure legends was an oversight in our graphing. The Y-axis in Fig 1 represents the cumulative number of times that the item has been selected.

---

## [Decision Letter · Decision Letter 1]

18 Sep 2023

PONE-D-23-17586R1The relationships between screen exposure, parent-child interactions and comprehension in 8-month-old infants: The mediating role of shared viewing and parent-child conversationPLOS ONE

Dear Dr. Luo,

Thank you for submitting your manuscript to PLOS ONE. After careful consideration, we feel that it has merit but does not fully meet PLOS ONE’s publication criteria as it currently stands. Therefore, we invite you to submit a revised version of the manuscript that addresses the points raised during the review process.

ACADEMIC EDITOR: Please do this major revision.

We look forward to receiving your revised manuscript.

Kind regards,

Anastassia Zabrodskaja, Ph.D.

Academic Editor

PLOS ONE

Reviewers' comments:

Reviewer's Responses to Questions

**Comments to the Author**

1. If the authors have adequately addressed your comments raised in a previous round of review and you feel that this manuscript is now acceptable for publication, you may indicate that here to bypass the “Comments to the Author” section, enter your conflict of interest statement in the “Confidential to Editor” section, and submit your "Accept" recommendation.

Reviewer #1: (No Response)

Reviewer #2: (No Response)

2. Is the manuscript technically sound, and do the data support the conclusions?

Reviewer #1: No

Reviewer #2: Yes

3. Has the statistical analysis been performed appropriately and rigorously? 

Reviewer #1: No

Reviewer #2: Yes

4. Have the authors made all data underlying the findings in their manuscript fully available?

Reviewer #1: No

Reviewer #2: No

5. Is the manuscript presented in an intelligible fashion and written in standard English?

Reviewer #1: No

Reviewer #2: Yes

6. Review Comments to the Author

Reviewer #1: I thank the authors for revising and resubmitting the manuscript. Unfortunately, many of my methodological issues still stand, and going through the manuscript again has brought some new issues to my attention. Therefore, I believe that the manuscript should go through a substantial revision before being considered for publication. In the following, I will only highlight a couple of major issues in the manuscript.

I am still not sure what "screen exposure", "shared viewing", and "parent-child conversations" are. I assume these are constructs exctracted from the attached questionnaire, but there are too many unanswered questions. Who developed the instrument? Has the instrument been validated? How were the constructs generated — theoretically, based on factor analysis, etc.? How valid are these constructs? What are their psychometric properties? How are the composite scores generated — are they sum scores, mean scores, factor loadings etc.? This is the most important predictor in the paper, and we know virtually nothing about it. All these questions should be answered in the text. Attaching the questionnaire does not help in figuring out how the responses were processed, unfortunately.

On p. 6, the authors say: "The questionnaire covers the following main points: whether or not the infant has been exposed to an electronic screen, types of electronic screen device, content viewed, purpose of screen viewing, the age of children's first screen exposure, frequency of screen exposure, the caregiver’s behavior during viewing, the screen interactivity. Sample items of shared viewing and parent-child conversation". However, only three factors are extracted from the questionnaire. Why were some of the theoretically important constructs combined? Was it a data-driven decision?

Also, I have an issue with screen exposure being coded as a yes-no variable. The questionnaire seems to allow for a much more nuanced (continuous) coding of screen exposure. I appreciate the fact that screen exposure may differ across cultures, but I find it hard to believe that there are many infants that have NO interactions whatsoever with digital media, given how ubiquitous they are.

My other big concern relates to the outcome variable. No summary statistics for the PCDI are provided, but I checked myself and the mean comprehension score is around 40 words (SD = 27), which is comparable to CDI data from other languages. However, nothing seems to predict differences in PCDI scores except for parent-child interactions and screen exposure. There are no effects of gender, risk factors (which is not defined in detail), parental education, or household income. This is very unusual, given that all these variables are known predictors of language development already at such an early age. For instance, gender alone is known to explain around 8%/9% of the variance in vocabulary size; therefore, it is puzzling that a model including both background factors and data on parent-child interactions only accounts for around 5% of the variance. I am not saying that the data is incorrect, but it is hard to interpret a model in which many important predictors of language development do not show any effect. Are there other factors that have not been included in the analyses that may explain the variance that is left unexplained by the included variables?

Once these issues have been dealt with, it will be easier to interpret the results (it is not possible at the moment because there are too many unknowns).

Reviewer #2: Thanks for the revised version, the paper is now much clearer and in my view closer to be accepted. However I do have some remaining comments that I believe should be addressed.

1. Attrition. There is still no info in the paper re the attrition which I believe is necessary.

2. Ethics. Now ok.

3. Method. The answer is informative, but I would like some of it to be included in the paper as well, e.g., who completed the q-naire (couldn’t find it in the revised text).

4. Quality cntrl. Now ok

5. Validations. Now mostly ok but the text states“We referred to the Chinese version of the BPCIS, which was validated by Jingjing, to evaluate the quality of parent-child interactions”. A ref to Jingjing is missing (29) here.

6. Screen exposure. In your answer you write that the ref I provided (Barr et al., 2020) contributed to the discussion but I cannot see how and the reference is not added. So what do you mean with “contributed to the discussion”?

7- 10: All issues addressed.

7. PLOS authors have the option to publish the peer review history of their article (what does this mean?). If published, this will include your full peer review and any attached files.

Reviewer #1: No

Reviewer #2: No

---

## [Author Response · Author response to Decision Letter 1]

23 Oct 2023

Dear Editors and Reviewers:

Thank you for your letter and for the reviewers’ comments concerning our manuscript entitled “The relationships between screen exposure, parent-child interactions and comprehension in 8-month-old infants: The mediating role of shared viewing and parent-child conversation” (ID: PONE-D-23-17586). Those comments are all valuable and very helpful for revising and improving our paper, as well as the important guiding significance to our researches. We have studied comments carefully and have made correction which we hope meet with approval.

Reviewer #1: 

I thank the authors for revising and resubmitting the manuscript. Unfortunately, many of my methodological issues still stand, and going through the manuscript again has brought some new issues to my attention. Therefore, I believe that the manuscript should go through a substantial revision before being considered for publication. In the following, I will only highlight a couple of major issues in the manuscript.

I am still not sure what "screen exposure", "shared viewing", and "parent-child conversations" are. I assume these are constructs exctracted from the attached questionnaire, but there are too many unanswered questions. Who developed the instrument? Has the instrument been validated? How were the constructs generated — theoretically, based on factor analysis, etc.? How valid are these constructs? What are their psychometric properties? How are the composite scores generated — are they sum scores, mean scores, factor loadings etc.? This is the most important predictor in the paper, and we know virtually nothing about it. All these questions should be answered in the text. Attaching the questionnaire does not help in figuring out how the responses were processed, unfortunately.

On p. 6, the authors say: "The questionnaire covers the following main points: whether or not the infant has been exposed to an electronic screen, types of electronic screen device, content viewed, purpose of screen viewing, the age of children's first screen exposure, frequency of screen exposure, the caregiver’s behavior during viewing, the screen interactivity. Sample items of shared viewing and parent-child conversation". However, only three factors are extracted from the questionnaire. Why were some of the theoretically important constructs combined? Was it a data-driven decision?

Also, I have an issue with screen exposure being coded as a yes-no variable. The questionnaire seems to allow for a much more nuanced (continuous) coding of screen exposure. I appreciate the fact that screen exposure may differ across cultures, but I find it hard to believe that there are many infants that have NO interactions whatsoever with digital media, given how ubiquitous they are.

My other big concern relates to the outcome variable. No summary statistics for the PCDI are provided, but I checked myself and the mean comprehension score is around 40 words (SD = 27), which is comparable to CDI data from other languages. However, nothing seems to predict differences in PCDI scores except for parent-child interactions and screen exposure. There are no effects of gender, risk factors (which is not defined in detail), parental education, or household income. This is very unusual, given that all these variables are known predictors of language development already at such an early age. For instance, gender alone is known to explain around 8%/9% of the variance in vocabulary size; therefore, it is puzzling that a model including both background factors and data on parent-child interactions only accounts for around 5% of the variance. I am not saying that the data is incorrect, but it is hard to interpret a model in which many important predictors of language development do not show any effect. Are there other factors that have not been included in the analyses that may explain the variance that is left unexplained by the included variables?

Once these issues have been dealt with, it will be easier to interpret the results (it is not possible at the moment because there are too many unknowns).

Responds to the reviewer’s comments:

1.Thanks for your comments, we have modified the description of the screen exposure questionnaire in the Methods part. We prepared a screen exposure questionnaire based on the screen time recommendations by the American Academy of Pediatrics and the studies of Wu et al., 2016 and Klakk et al., 2020，The questionnaire consisted of four parts, and since the first and third parts of the questionnaire were multiple choice questions for descriptive statistics (The results are presented in Table 2 and Figure 1), we calculated Cronbach's alpha coefficients only for the parts 2 and 4. Regarding the question about the composite scores of the questionnaire, we did not calculate the composite scores, and the results for shared viewing, parent-child conversation, etc. were derived from a single item in the questionnaire. 

2. Thanks for your comments. These three factors were our main focus and the main purpose of this research (the role of co-viewing and parent-child conversation during co-viewing in children's screen exposure). Other results of the questionnaire are reported in Table 2 and Figure 1. The scores for these three factors were derived from individual items rather than composite scores.

3. Thanks for your comments. The results of dividing infant screen exposure into more levels of coding are reported in Table 2. Fewer infants in China are screen-exposed than in developed countries (Kabali et al., 2015; Chen et al., 2019) (about 50% of Chinese infants are exposed to electronic screens. Shan et al., 2023), and with reference to the method of Zhou et al., 2020; Chen et al., 2018 and Sun et al., 2020 who coded screen exposure as a yes/no variable. Based on the AAP guidelines on screen use, which state that electronic screens are not recommended for children under 2 years of age, and that children older than 2 years of age should have less than 2 hours of screen time per day. The children in this study were 8 months old and were therefore divided into a screen-exposed group and a non-screen-exposed group. Our screen exposure questionnaire was not designed to take into account the screen as a backdrop (e.g., whether children watch television, rather than whether television is used as a backdrop when children play), so there are many infants that have NO interactions whatsoever with digital media. Chinese society and culture maintains an attitude that screen exposure hampers early childhood development (LI Min-yi et al., 2022), so some parents try to avoid exposing their children to electronic screens. According to the results of previous studies, the proportion of Chinese infants exposed to screens and the duration of exposure is lower than that of infants in developed countries such as the United States and Singapore (Li et al., 2020 - The Relationships between Screen Use and Health Indicators among Infants, Toddlers, and Preschoolers: A Meta-Analysis and Systematic Review). 

4. Thank you very much for this important comments. Since the R2 in the hierarchical multiple regression is only 5%，similarly, we considered that there might be other predictors of language development (e.g. a rich language environment at home) that were not included in the model, but the background factors we studied only those components that were included in the model. And a small R2-value may blur the contribution of background factors such as gender, parental education, household income, risk factors, etc., to PCDI. And the relatively small sample size is also considered to be a possible reason for the small contribution of other background factors. In searching for relevant studies, we found that the contribution of gender to infants' language development is still under discussion (Oller et al., 2023; Hyde & Linn 1998-Gender Differences in Verbal Ability: A Meta-Analysis. Psychological Bulletin). Some studies have found that gender contributes to language development (Shen & wang, 2018; Cao et al., 2019;), but others have found that gender has no significant effect on language development (Hao et al., 2005; Wang et al., 2019; Oller et al., 2023). A Chinese study found that although parental education has a significant effect on infant language, the relationship between maternal education and infant language development is no longer significant after controlling for parenting knowledge and parenting behaviour (Wang et al., 2019). And Bian et al., 2007 found that children in families with moderate social-economic status had the highest levels of language development. (Possibly influenced by China's social and economic environment, where middle-income groups take longer breaks and spend more time with children, promoting children's language development.)

Reviewer #2:

Thanks for the revised version, the paper is now much clearer and in my view closer to be accepted. However I do have some remaining comments that I believe should be addressed.

1. Attrition. There is still no info in the paper re the attrition which I believe is necessary.

2. Ethics. Now ok.

3. Method. The answer is informative, but I would like some of it to be included in the paper as well, e.g., who completed the q-naire (couldn’t find it in the revised text).

4. Quality cntrl. Now ok

5. Validations. Now mostly ok but the text states “We referred to the Chinese version of the BPCIS, which was validated by Jingjing, to evaluate the quality of parent-child interactions”. A ref to Jingjing is missing (29) here.

6. Screen exposure. In your answer you write that the ref I provided (Barr et al., 2020) contributed to the discussion but I cannot see how and the reference is not added. So what do you mean with “contributed to the discussion”?

7- 10: All issues addressed.

Responds to the reviewer’s comments:

1. Thanks for your comments, we've added information about attrition to the methods section.

2. Thank you for this comment.

3. Thanks to your comments, we have included relevant information such as who completed the questionnaire in the quality control section.

4. Thank you for this comment.

5.Thanks to your comments, we have added a reference to Jingjing.

6. Thank you for the reminder and we apologize for the error we made, we have revised the limitations rather than the discussion based on the Barr et al. study. We have added references to the fourth point of the limitations. Your comments are very helpful.

---

## [Decision Letter · Decision Letter 2]

12 Nov 2023

PONE-D-23-17586R2The relationships between screen exposure, parent-child interactions and comprehension in 8-month-old infants: The mediating role of shared viewing and parent-child conversationPLOS ONE

Dear Dr. Luo,

Thank you for submitting your manuscript to PLOS ONE. After careful consideration, we feel that it has merit but does not fully meet PLOS ONE’s publication criteria as it currently stands. Therefore, we invite you to submit a revised version of the manuscript that addresses the points raised during the review process.

Please revise according to the reviewers' comments.

We look forward to receiving your revised manuscript.

Kind regards,

Anastassia Zabrodskaja, Ph.D.

Academic Editor

PLOS ONE

Reviewers' comments:

Reviewer's Responses to Questions

**Comments to the Author**

1. If the authors have adequately addressed your comments raised in a previous round of review and you feel that this manuscript is now acceptable for publication, you may indicate that here to bypass the “Comments to the Author” section, enter your conflict of interest statement in the “Confidential to Editor” section, and submit your "Accept" recommendation.

Reviewer #1: (No Response)

Reviewer #2: All comments have been addressed

2. Is the manuscript technically sound, and do the data support the conclusions?

Reviewer #1: Partly

Reviewer #2: Yes

3. Has the statistical analysis been performed appropriately and rigorously? 

Reviewer #1: Yes

Reviewer #2: Yes

4. Have the authors made all data underlying the findings in their manuscript fully available?

Reviewer #1: Yes

Reviewer #2: Yes

5. Is the manuscript presented in an intelligible fashion and written in standard English?

Reviewer #1: No

Reviewer #2: Yes

6. Review Comments to the Author

Reviewer #1: The authors answered some of my questions and have made a relatively good case in #4 about why eg. education and child gender do not predict vocabulary development in their model. However, many of my questions are still unanswered.

Reviewer #2: All of my questions have been fully answered and I the paper is publishable. I congratulate the authors to a nice paper.

7. PLOS authors have the option to publish the peer review history of their article (what does this mean?). If published, this will include your full peer review and any attached files.

Reviewer #1: No

Reviewer #2: No

---

## [Author Response · Author response to Decision Letter 2]

6 Dec 2023

Dear Editors and Reviewers:

Thank you for your letter and for the reviewers’ comments concerning our manuscript entitled “The relationships between screen exposure, parent-child interactions and comprehension in 8-month-old infants: The mediating role of shared viewing and parent-child conversation” (ID: PONE-D-23-17586). We have studied comments carefully and have made correction which we hope meet with approval. Revised portion are marked in yellow in the paper. 

Firstly, we would like to express my sincere gratitude for reviewer #2, the comments and suggestions of reviewer #2 have greatly improved the quality of the manuscript. Furthermore, we would also like to extend my heartfelt thanks for the recognition and positive feedback of reviewer #2 on the manuscript.

Reviewer #1 noted in his comment that many of his questions remained unanswered, possibly because we revised the manuscript in accordance with Reviewer #1's comments without providing specific answers to the related questions. Therefore, we have reviewed reviewer #1's comments from the two previous revisions and provided specific answers in this response.

Reviewer #1: 

The authors answered some of my questions and have made a relatively good case in #4 about why eg. education and child gender do not predict vocabulary development in their model. However, many of my questions are still unanswered.

Responds to the reviewer’s comments:

Thank you for reviewing our manuscript and providing valuable feedback and comments. Your previous comments and suggestions have been very helpful in our revisions. We have re-evaluated your previous comments and provided specific responses. We greatly appreciate your patient guidance, and your input is crucial to our work.

We are very sorry for our negligence in answering the questions relating to the screen exposure questionnaire. In the methods section of the manuscript, we have described the constructs of "screen exposure," "shared viewing," and "parent-child conversation.", the reliability of the questionnaire, the psychometric properties and the composite scores of the questionnaire. In this response, we have answered those questions according to your comments.

We developed the screen exposure questionnaire based on the screen time recommendations by the American Academy of Pediatrics and the studies of Wu et al. (2016) [1] and Klakk et al. (2020) [2]. Regarding the issue of whether this instrument has been validated, we have calculated the Cronbach's alpha coefficient (0.896 - 0.912). The questionnaire consisted of 11 items, 4 of which were analyzed by descriptive statistics (Fig 2). Many of the questionnaires used in previous studies [3-15] on screen exposure were not validated for reliability, and psychological characteristics were obtained from individual items without calculating composite scores. The screen exposure questionnaire used by Çelen Yoldas T et al. (2021) [16], Shah PE et al. (2021) [17], and Lin Y, et al. (2022) [18], which were referenced in this study, included items on co-viewing and parent-child conversation during screen exposure.

The item in the screen exposure questionnaire used in the present study regarding whether or not the children had been exposed to screens was informed by the study of Zhou et al. [19] (Has your child been exposed to an electronic screen device?). The "shared viewing" item was informed by the study of Zimmerman FJ et al. [20], in this study, parents were asked "How often do you watch electronic screen devices with your children when they are exposed to them?" Similarly, Zimmerman FJ et al. used a single item to evaluate this psychometric property. The "parent-child conversations" item was informed by the study of Shah PE et al. [21] In the present study, parents were asked "When you and your child watch an electronic screen device together, how often do you talk with your child about the screen contents?" Similarly, Shah PE et al. used a single item to evaluate this psychometric property and there were only two items on screen exposure in Shah PE et al. study: Hours of television viewing and Parent conversation during shared television viewing. As in previous studies [22-28], the present study evaluated the psychometric properties of the screen exposure questionnaire with individual items, and composite scores were not calculated.

Response to comment: However, only three factors are extracted from the questionnaire. Why were some of the theoretically important constructs combined? Was it a data-driven decision?"

 Response: We have carefully reviewed the comments. Firstly, the other factors were analyzed by descriptive statistics (see Table 1 and Figure 1). The three factors you mentioned are the ones we intended to explore during the research design phase of this study. We wanted to explore how parental behaviors (co-viewing and talking about screen contents) during infants' exposure to screens had an impact on children's development, and were therefore not data-driven.

Response to comment: Why was screen exposure coded as a yes-no variable？

Response: We have coded the screen exposure as a yes-no variable according to the American Academy of Pediatrics' screen time guidelines (AAP calls for no screen time at all for children until 18 to 24 months), The subjects in studies of Wu et al. [29] and Tombeau Cost K et al. [30] were similarly younger than 18 months, and they also coded screen exposure as a yes-no variable.

Response to comment: it hard to believe that there are many infants that have NO interactions whatsoever with digital media, given how ubiquitous they are. 

Response: The screen exposure questionnaire in the present study was not designed to take into account the screen turned on in the background, Therefore, our findings demonstrated that there were many infants that have NO interactions whatsoever with digital media. In Chinese society and culture, attitudes towards electronic screen devices are relatively negative, especially for children's development [31], Therefore, some Chinese parents try to avoid exposing their children to electronic screen devices. According to previous research, compared to developed countries like the United States and Singapore, Chinese infants and young children have lower rates of screen exposure and spend less time using screens [32]. Regarding the proportion of infants and toddlers exposed to screens, our results are similar to findings in previous studies that conducted in China [33,34]. With respect to the social and cultural differences between China and other developed countries like the United States and those in the European Union, we therefore again answer this important comment. 

Regarding issues related to the BPCIS, we noted in the methods section that we have used the Chinese version of the BPCIS revised by Jingjing Lai. Furthermore, we have revised the methods section of the manuscript based on Jingjing Lai's thesis. 

We have tried our best to improve the language in the manuscript and have modified some confusing sentences, making them concise and easy to read. We would greatly appreciate it if you could kindly note any other errors so that we may revise our manuscript accordingly based on your valuable suggestions.

If you have any further questions, please do not hesitate to inform us.

Reviewer #2: 

All of my questions have been fully answered and I the paper is publishable. I congratulate the authors to a nice paper.

Responds to the reviewer’s comments:

We would like to extend my sincere gratitude once again for your endorsement of our manuscript. Your recognition and support are deeply appreciated.

References

1. Wu XY, Tao SM, Zhang SC, Zhang YK, Huang K, Tao FB. [Analysis on risk factors of screen time among Chinese primary and middle school students in 12 provinces]. Zhonghua Yu Fang Yi Xue Za Zhi. 2016 Jun;50(6):508-13. Chinese. doi: 10.3760/cma.j.issn.0253-9624.2016.06.007. PMID: 27256730.

2. Klakk H, Wester CT, Olesen LG, Rasmussen MG, Kristensen PL, Pedersen J, Grøntved A. The development of a questionnaire to assess leisure time screen-based media use and its proximal correlates in children (SCREENS-Q). BMC Public Health. 2020 May 12;20(1):664. doi: 10.1186/s12889-020-08810-6. PMID: 32397984; PMCID: PMC7216486.

3. He Hui, Wei Zhuang, Liu Chunyang, Liang Aimin. Survey on Children's Use of Electronic Products in Beijing. Maternal and Child Health Care of China. January. 2016;31(2):351–353. doi:10. 7620 /zgfybj. j. issn. 1001－4411. 2016. 02. 47

4. ZHOU Shan-shan, YAN Shuang-qin, CAO Hui, GAO Guo-peng, CAI Zhi-ling, GU Chun-li, LIU Ting-ting, QIAN Zhi-kan, WANG Hao, LIU Zi-jian, TAO Fang-biao. Prevalence and the risk factors of television viewing by infants and toddlers in Ma′anshan city. Chinese Journal of Child Health Care, 2020, 28(1): 61-64.

5. FANG Hong-ying, TANG Yin-xia, XU Cheng-heng,YANG Fei-fei,ChEN Shang-hui. Survey on exposure to electronic devices of preschool children in Tongling city. Chinese Journal of Child Health Care, 2018, 26(8): 913-915.

6. YEYi, ZHOU Yong hai, CHEN Ke. Television and DVD/VCD exposure in children younger than three years old in Wenzhou city. Chinese preventive medicine , 2009, 10(12):1060–1064.

7. WANG Xuena, DU Wenwen, ZHANG Meng, et al. Investigation on screen time and psychosocial problems of preschoolers during the prevalence of COVID—19. China Journal of Health Psychology, 2021;29(4):564–568.

8. SHAN Ruijie, HAN Jing, QU Keli, YUE Lei, CUI Naixue. Associated factors of screen exposure in infants. Chinese Journal of Child Health Care, 2023, 31(4): 374-378.

9. Zhao Shanshan, Yao Conghua, Xu Jinping. Prospective cohort study and risk factor analysis of language delay based on outpatient in Xiamen.Chin J Appl Clin Pediatr, 2021,36(14):1094-1097.

10. Tombeau Cost K, Korczak D, Charach A, Birken C, Maguire JL, Parkin PC, Szatmari P. Association of Parental and Contextual Stressors With Child Screen Exposure and Child Screen Exposure Combined With Feeding. JAMA Netw Open. 2020 Feb 5;3(2):e1920557. doi: 10.1001/jamanetworkopen.2019.20557. PMID: 32022883. 

11. Sundqvist A, Koch FS, Birberg Thornberg U, Barr R, Heimann M. Growing Up in a Digital World - Digital Media and the Association With the Child's Language Development at Two Years of Age. Front Psychol. 2021 Mar 18;12:569920. doi: 10.3389/fpsyg.2021.569920. PMID: 33815187; PMCID: PMC8015860.

12. Taylor G , Monaghan P , Westermann G .Investigating the association between children's screen media exposure and vocabulary size in the UK.Routledge, 2018(1).DOI:10.1080/17482798.2017.1365737. 

13. Çelen Yoldaş T, Özmert EN. Communicative Environmental Factors Including Maternal Depression and Media Usage Patterns on Early Language Development. Matern Child Health J. 2021 Jun;25(6):900-908. doi: 10.1007/s10995-021-03125-3. Epub 2021 Apr 27. PMID: 33905063. 

14. Kabali HK, Irigoyen MM, Nunez-Davis R, Budacki JG, Mohanty SH, Leister KP, Bonner RL Jr. Exposure and Use of Mobile Media Devices by Young Children. Pediatrics. 2015 Dec;136(6):1044-50. doi: 10.1542/peds.2015-2151. Epub 2015 Nov 2. PMID: 26527548.

15. Shah PE, Hirsh-Pasek K, Kashdan TB, Harrison K, Rosenblum K, Weeks HM, Singh P, Kaciroti N. Daily television exposure, parent conversation during shared television viewing and socioeconomic status: Associations with curiosity at kindergarten. PLoS One. 2021 Oct 28;16(10):e0258572. doi: 10.1371/journal.pone.0258572

16. Çelen Yoldaş T, Özmert EN. Communicative Environmental Factors Including Maternal Depression and Media Usage Patterns on Early Language Development. Matern Child Health J. 2021 Jun;25(6):900-908. doi: 10.1007/s10995-021-03125-3. Epub 2021 Apr 27. PMID: 33905063. 

17. Shah PE, Hirsh-Pasek K, Kashdan TB, Harrison K, Rosenblum K, Weeks HM, Singh P, Kaciroti N. Daily television exposure, parent conversation during shared television viewing and socioeconomic status: Associations with curiosity at kindergarten. PLoS One. 2021 Oct 28;16(10):e0258572. doi: 10.1371/journal.pone.0258572

18. Lin Y, Zhang X, Huang Y, Jia Z, Chen J, Hou W, Zhao L, Wang G, Zhu J. Relationships between screen viewing and sleep quality for infants and toddlers in China: A cross-sectional study. Front Pediatr. 2022 Oct 10;10:987523. doi: 10.3389/fped.2022.987523. PMID: 36299700; PMCID: PMC9589267.

19. Zhou SS, Yan SQ, Cao H et al. Prevalence and the risk factors of television viewing by infants and toddlers in Ma′anshan city. Chin J Child Health Care. 2020 Sep;28(1):61-64. doi:10.11852/zgetbjzz2019-1087

20. Zimmerman FJ, Christakis DA, Meltzoff AN. Associations between media viewing and language development in children under age 2 years. J Pediatr. 2007 Oct;151(4):364-8. doi: 10.1016/j.jpeds.2007.04.071. Epub 2007 Aug 7. PMID: 17889070. 

21. Shah PE, Hirsh-Pasek K, Kashdan TB, Harrison K, Rosenblum K, Weeks HM, Singh P, Kaciroti N. Daily television exposure, parent conversation during shared television viewing and socioeconomic status: Associations with curiosity at kindergarten. PLoS One. 2021 Oct 28;16(10):e0258572. doi: 10.1371/journal.pone.0258572

22. He Hui, Wei Zhuang, Liu Chunyang, Liang Aimin. Survey on Children's Use of Electronic Products in Beijing. Maternal and Child Health Care of China. January. 2016;31(2):351–353. doi:10. 7620 /zgfybj. j. issn. 1001－4411. 2016. 02. 47

23. FANG Hong-ying, TANG Yin-xia, XU Cheng-heng,YANG Fei-fei,ChEN Shang-hui. Survey on exposure to electronic devices of preschool children in Tongling city. Chinese Journal of Child Health Care, 2018, 26(8): 913-915.

24. WANG Xuena, DU Wenwen, ZHANG Meng, et al. Investigation on screen time and psychosocial problems of preschoolers during the prevalence of COVID—19. China Journal of Health Psychology, 2021;29(4):564–568.

25. SONG Wenya, PAN Xiaowei, SUN Longxia, NI Yu. Effects of screen exposure on cognitive-social development in preschooler.Chines Journal of Woman and Child Health Reserach, 2021;32(10):1452–1457.

26. Shah PE, Hirsh-Pasek K, Kashdan TB, Harrison K, Rosenblum K, Weeks HM, Singh P, Kaciroti N. Daily television exposure, parent conversation during shared television viewing and socioeconomic status: Associations with curiosity at kindergarten. PLoS One. 2021 Oct 28;16(10):e0258572. doi: 10.1371/journal.pone.0258572

27. Taylor G, Monaghan P, Westermann G. Investigating the association between children’s screen media exposure and vocabulary size in the UK. Journal of Children and Media. 2018 Jan 2;12(1):51–65.

28. Takahashi I, Obara T, Ishikuro M, Murakami K, Ueno F, Noda A, Onuma T, Shinoda G, Nishimura T, Tsuchiya KJ, Kuriyama S. Screen Time at Age 1 Year and Communication and Problem-Solving Developmental Delay at 2 and 4 Years. JAMA Pediatr. 2023 Oct 1;177(10):1039-1046. doi: 10.1001/jamapediatrics.2023.3057.

29. Wu XY, Tao SM, Zhang SC, Zhang YK, Huang K, Tao FB. [Analysis on risk factors of screen time among Chinese primary and middle school students in 12 provinces]. Zhonghua Yu Fang Yi Xue Za Zhi. 2016 Jun;50(6):508-13. Chinese. doi: 10.3760/cma.j.issn.0253-9624.2016.06.007. PMID: 27256730.

30. Tombeau Cost K, Korczak D, Charach A, Birken C, Maguire JL, Parkin PC, Szatmari P. Association of Parental and Contextual Stressors With Child Screen Exposure and Child Screen Exposure Combined With Feeding. JAMA Netw Open. 2020 Feb 5;3(2):e1920557.

31. WANG Wei,WEI Yanan. A Survey Study on Parents' Attitudes toward Young Children's Use of Electronic Products--Taking Chongqing Municipality as an Example. Education Guide (the second half of the month),2017(2):75-78.

32. Li, C. , Cheng, G. , Sha, T. , Cheng, W. , & Yan, Y. . (2020). The relationships between screen use and health indicators among infants, toddlers, and preschoolers: a meta-analysis and systematic review. International Journal of Environmental Research and Public Health, 17(19), 7324.

33. Zhou SS, Yan SQ, Cao H et al. Prevalence and the risk factors of television viewing by infants and toddlers in Ma′anshan city. Chin J Child Health Care. 2020 Sep;28(1):61-64. doi:10.11852/zgetbjzz2019-1087

34. SHAN Ruijie, HAN Jing, QU Keli, YUE Lei, CUI Naixue. Associated factors of screen exposure in infants. Chinese Journal of Child Health Care, 2023, 31(4): 374-378.

---

## [Decision Letter · Decision Letter 3]

11 Dec 2023

The relationships between screen exposure, parent-child interactions and comprehension in 8-month-old infants: The mediating role of shared viewing and parent-child conversation

PONE-D-23-17586R3

Dear Dr. Luo,

We’re pleased to inform you that your manuscript has been judged scientifically suitable for publication and will be formally accepted for publication once it meets all outstanding technical requirements.

Kind regards,

Anastassia Zabrodskaja, Ph.D.

Academic Editor

PLOS ONE

Additional Editor Comments (optional):

Reviewers' comments:

Reviewer's Responses to Questions

**Comments to the Author**

1. If the authors have adequately addressed your comments raised in a previous round of review and you feel that this manuscript is now acceptable for publication, you may indicate that here to bypass the “Comments to the Author” section, enter your conflict of interest statement in the “Confidential to Editor” section, and submit your "Accept" recommendation.

Reviewer #2: All comments have been addressed

2. Is the manuscript technically sound, and do the data support the conclusions?

Reviewer #2: Yes

3. Has the statistical analysis been performed appropriately and rigorously? 

Reviewer #2: Yes

4. Have the authors made all data underlying the findings in their manuscript fully available?

Reviewer #2: Yes

5. Is the manuscript presented in an intelligible fashion and written in standard English?

Reviewer #2: Yes

6. Review Comments to the Author

Reviewer #2: (No Response)

7. PLOS authors have the option to publish the peer review history of their article (what does this mean?). If published, this will include your full peer review and any attached files.

Reviewer #2: No

---

## [Editor Report · Acceptance letter]

20 Dec 2023

PONE-D-23-17586R3 

PLOS ONE

Dear Dr. Luo, 

I'm pleased to inform you that your manuscript has been deemed suitable for publication in PLOS ONE. Congratulations! Your manuscript is now being handed over to our production team.

Kind regards, 

on behalf of

Professor Anastassia Zabrodskaja 

Academic Editor

PLOS ONE